# From Spheroids to Organoids: The Next Generation of Model Systems of Human Cardiac Regeneration in a Dish

**DOI:** 10.3390/ijms222413180

**Published:** 2021-12-07

**Authors:** Mariangela Scalise, Fabiola Marino, Luca Salerno, Eleonora Cianflone, Claudia Molinaro, Nadia Salerno, Antonella De Angelis, Giuseppe Viglietto, Konrad Urbanek, Daniele Torella

**Affiliations:** 1Department of Experimental and Clinical Medicine, University Magna Græcia of Catanzaro, 88100 Catanzaro, Italy; m.scalise@unicz.it (M.S.); marino@unicz.it (F.M.); l.salerno@unicz.it (L.S.); viglietto@unicz.it (G.V.); urbanek@unicz.it (K.U.); 2Department of Medical and Surgical Sciences, University Magna Græcia of Catanzaro, 88100 Catanzaro, Italy; cianflone@unicz.it (E.C.); c.molinaro@unicz.it (C.M.); nadia.salerno@unicz.it (N.S.); 3Department of Experimental Medicine, University of Campania “L. Vanvitelli”, 80138 Naples, Italy; antonella.deangelis@unicampania.it

**Keywords:** organoid, pluripotent stem cell, adult stem cell, cardiac stem cell, cardioid, heart regeneration

## Abstract

Organoids are tiny, self-organized, three-dimensional tissue cultures that are derived from the differentiation of stem cells. The growing interest in the use of organoids arises from their ability to mimic the biology and physiology of specific tissue structures in vitro. Organoids indeed represent promising systems for the in vitro modeling of tissue morphogenesis and organogenesis, regenerative medicine and tissue engineering, drug therapy testing, toxicology screening, and disease modeling. Although 2D cell cultures have been used for more than 50 years, even for their simplicity and low-cost maintenance, recent years have witnessed a steep rise in the availability of organoid model systems. Exploiting the ability of cells to re-aggregate and reconstruct the original architecture of an organ makes it possible to overcome many limitations of 2D cell culture systems. In vitro replication of the cellular micro-environment of a specific tissue leads to reproducing the molecular, biochemical, and biomechanical mechanisms that directly influence cell behavior and fate within that specific tissue. Lineage-specific self-organizing organoids have now been generated for many organs. Currently, growing cardiac organoid (cardioids) from pluripotent stem cells and cardiac stem/progenitor cells remains an open challenge due to the complexity of the spreading, differentiation, and migration of cardiac muscle and vascular layers. Here, we summarize the evolution of biological model systems from the generation of 2D spheroids to 3D organoids by focusing on the generation of cardioids based on the currently available laboratory technologies and outline their high potential for cardiovascular research.

## 1. Defining an Organoid

The term “organoid” refers to mini clusters of growing cells able to self-organize in vitro and differentiate into functional cell types, resembling an organ 3D structure and function. The word “organoid” is mainly used to describe such structure derived from stem cells. In multicellular organisms, stem cells are undifferentiated or partially committed cells that can differentiate into various types of cells and proliferate indefinitely to produce more of the same stem cell (the so-called self-renewal). Stem cells are present in both embryonic and adult organisms, but they have slightly different properties in each. In general, they can generate all tissues of the developing embryo and maintain tissue homeostasis in adults. In the past decade, these prototypical stem cell features have been exploited to develop the organoids in vitro. Organoids are therefore stem cell-derived and self-organizing 3D cultures that phenocopy cell-type composition, architecture, and, to a certain extent, functionality of different tissues [1]. The developmental potential of the initiating stem cells influences how complex the organoid can be. The organoids resemble specific features of organs in vivo: an organoid must indeed contain more than one cell type of the organ it models; it should exhibit some function related to that organ; and the cells should be organized similarly to the tissue of the organ. Additionally, organoids’ formation recapitulates characteristic processes of self-organization during development [2]. However, although most organoid cultures develop functional tissue units, they lack elements such as mesenchymal, stromal, immune, and neural cells that populate tissue in vivo. Failing to recapitulate the complexity of native organs, the (partial) absence of a mesenchymal compartment, vascularization, and microbiome represents therefore a limit of organoid technology. Yet, recent studies are dealing with trying to overcome this limitation through obtaining a tridimensional structure that more closely reproduces the whole cellular diversity of the tissue microenvironment [3].

Although still imperfect, organoids represent an attractive model for studying human biology and disease, carrying the potential to answer several unresolved questions. Organoids rely on artificial extracellular matrices (ECM) to facilitate their self-organization into structures that resemble native tissue. ECM components such as laminin, fibronectin and collagen engage the integrin receptors and support to maintain cell identity and function. Organoids are similar to primary tissue in their composition and architecture, harboring small populations of genomically stable, self-renewing stem cells that give rise to fully differentiated progeny comprising all major cell lineages. Among the advantages of their use, organoids can be expanded indefinitely, cryopreserved as biobanks, and easily manipulated using techniques such as those established for traditional 2D monolayer culture. The study of organoid formation can provide valuable information about the mechanisms underlying development and organ regeneration, underscoring their value for basic biological research in addition to their potential application in drug testing and molecular medicine. The potential of organoids to complement existing model systems and extend basic research and drug discovery is becoming more widely appreciated [4,5].

In this review, we discuss the evolution of biological model systems tracing the main methodologies that led us from 2D- to the 3D-cell cultures and organoids’ generation. In particular, we thoroughly assess the current methodologies used to generate cardiac spheroids and 3D-cell structures derived from stem cells, and highlight the potential of cardiac organoids (“cardioids”) as a model system for the understanding of heart development and for the study of human cardiac regeneration in a dish. Furthermore, we argue that cardioids are ideal human preclinical models, useful to simulate pathological processes as well as to test drug toxicity, highlighting their current limitations that remain to be addressed.

## 2. A Brief Historical Perspective of Organoid Development

The first effort in describing in vitro self-organization and differentiation goes back in time to the beginning of the 20th century, when Wilson showed that dissociated sponge cells, when kept under appropriate conditions, degenerate, resulting in small masses of undifferentiated tissue able to grow and differentiate into complete sponges [6] (Figure 1). Some decades later, Holtfreter performed dissociation–reaggregation experiments with dissociated amphibian pro-nephrons [7] (Figure 1). In 1960, Weiss and Taylor conducted the same experiments with different organs from chicken embryos [8], and shortly later, Pierce and Verney described the differentiation of embryoid bodies (EBs) in vitro [9] (Figure 1). At the same time, Steinberg proposed a differential adhesion hypothesis according to which thermodynamic effects regulate the cell sorting and rearrangement in surface adhesion [10] (Figure 1). Stem cell research flourished when different groups isolated pluripotent stem cells from mouse embryos [11,12] (Figure 1). Subsequently, other groups focused on improving cell culture protocols to mimic the in vivo environment conditions. Li et al. showed that breast epithelium organizes into 3D ducts and increases milk protein secretion when grown in a tumor extracellular matrix [13] (Figure 1). Jennings et al. demonstrated similar structures in alveolar type II epithelial cells [14] (Figure 1). An important event in this field occurred in 1998 when Thompson et al. isolated and cultured the first embryonic stem cell line from human blastocysts [15] (Figure 1). The watershed from 2D to spheroid and organoid culture (3D) occurred with the generation of 3D cerebral cortex tissue from pluripotent stem cells by Eiraku et al. [16] (Figure 1). Successively, intestinal 3D structures were generated from adult intestinal stem cells in Matrigel [17], kicking off various works in other systems, including stomach, liver, pancreas, lungs, kidney, brain, and retina [18,19,20,21,22,23,24] (Figure 1). Moreover, in the recent severe acute respiratory syndrome coronavirus 2 (SARS-CoV-2) pandemic, to elucidate how the virus can damage lungs, liver, and kidney tissues, causing some of the severe complications seen in patients with coronavirus disease—2019 (COVID-19), several researchers have focused on organoids that could also facilitate screening for potential new drugs. Bronchial organoids with four distinct cell types, made from frozen cells from the bronchi’s epithelium—the outer cell layer—have been developed [25] (Figure 1). By infecting these organoids with SARS-CoV-2, it was established that the virus primarily targets stem cells, or basal cells, that supply the epithelium. At the same time, it does not easily penetrate the protective secretory cells, the Clara cells [26]. Human blood vessel organoids have been recently derived from human pluripotent stem cells (hPSCs) for modeling and identifying the regulators of diabetic vasculopathy [27] (Figure 1). Finally, very recently, a small three-dimensional model of the heart was obtained from stem cells in vitro. Heart organoids can reproduce specific functions of a heart chamber and of pathological conditions such as congenital heart defects and tissue damage after a heart attack [28] (Figure 1).

## 3. The Basis of 3D Cellular Structure Formation

Understanding the principles that lead to organization during tissue development is fundamental to developmental biology. Cell sorting is the process by which cohering disorganized aggregates of cells establish structured tissues. These cellular aggregates from disorganized structures become homogeneous tissue domains [29]. The ability of disordered cell aggregates to restore normal tissue architecture suggests that understanding the mechanisms underlying cell sorting in vivo should prove instructive in understanding the processes that govern and stabilize the definitive relationships of the tissues associated with each other in the various organs. These cells can sometimes further divide to give rise to more differentiated progeny, which is further displaced.

The basis of this organ self-assembly seems to arise from the segregation of cells with similar adhesive properties into domains that achieve the most thermodynamically stable pattern, known as Steinberg’s differential adhesion hypothesis [30]. Differential adhesion is mediated by cell surface adhesion proteins, for example, in separating vertebrate neural and epidermal ectoderm [31,32], where differential epithelial and neural and cadherin expression mediates cell sorting out. The events of cell sorting do not mimic the pathways of the morphogenic movements, whereas during normal morphogenesis, the tissues of single organs do not sort out into their final form from random mixtures of the constituent cells. Despite the latter, an understanding of the mechanisms by which cultured heterotypic cell aggregates generate patterned arrays of tissues shows promise for analyzing the processes that the embryo employs to produce the definitive organization of tissues brought into association by prior morphogenetic cell movements.

A second mechanism that can influence tissue morphogenesis is the correct and spatially restricted progenitor cell fate decisions. Progenitor cells give rise to more differentiated progeny, which, because of spatial constraints of the tissue and division orientation, are forced into a more superficial position that promotes their differentiation. This stratification depends upon proper stem cell division orientation, the interplay of symmetric and asymmetric divisions, and the migration of differentiated daughter cells to defined locations within the tissue [33,34]. A common technique used in studying tissue organization is re-aggregation, which breaks the tissue down into its simplest components, the cells, and allows them to re-aggregate in a simplified environment, devoid of surrounding tissues. It enables the investigator to observe how these cells interact to form the tissue and manipulate the components to determine which are essential in this process. Re-aggregation studies have been used since the early 20th century, first using simple organisms such as sea urchins and sponges and later using more complex, multi-layered tissues such as limb buds and retina from chicks [35,36]. These studies have revealed the innate ability of these multi-layered tissues to self-organize in vitro in the absence of many extrinsic cues and scaffolds. Stem cells (SCs), cultured in a dish, proliferate as a monolayer in two-dimensions (2D) and frequently require indeterminate or xenogeneic materials, including attachment substrates, cytokines, growth factors, as well as serum, to be effectively maintained and expanded in vitro. Xenogeneic contaminants from any non-human feeder cells or foreign components of the culture system hinder clinical application [37]. Monolayer culture requires interaction to maintain self-renewal and potency of cells, cell differentiation, vitality, expression of genes and proteins, responsiveness to stimuli, drug metabolism, and other cellular functions, which are highly inefficient for large-scale expansion of cells [38,39]. In particular, 2D attachment influences cell shape and structure [40], leading to cell flattening and changes in the internal cytoskeleton and nuclear shape [41], which in turn induces gene and protein expression changes [42,43]. After isolation from the tissue and transfer to the 2D conditions, cells start to lose their morphology, affecting their function [44], the organization of the structures inside the cell, secretion, and cell signaling [45]. Further studies have shown that the composition and organization of the ECM can also send biochemical and mechanical signals for cell differentiation [46]. Two-dimensional culture techniques and applications have been practiced for most primary and established cell lines and standardized for analytical assays ranging from microscopy and counting cells to the study of disease processes and drug testing [47]. When in 2D, cells have more surface area in contact with the plastic and culture media than with other cells [48], forcing them into a polarization that does not reflect physiological conditions. Two-dimensional culture has been used to differentiate SCs into many specialized cells, including chondrocytes, osteocytes, adipocytes, cardiomyocytes, smooth muscle cells, and hepatocytes [49,50]. One of the advantages of monolayer culture is that it allows for uniform treatment for the differentiation of cells [50]. In some cases, 2D cultures, however, result in a lack of resulting functional derivatives [51]. Further studies have shown that 2D monolayer culture fails to reproduce animal physiology [52] appropriately and is insufficient to validate drug discovery [53]. Cultures of pluripotent stem cells (PSCs) on dishes are coated with ECM components (such as Matrigel, laminin, collagen, or gelatin) and mouse embryonic fibroblasts (MEFs) feeder layer to support attachment [54]. Two-dimensional expansion of embryonic stem cells (ESCs) has been enhanced using completely defined xeno-free culture media and attachment substrates such as albumin-free E8 media and humanized recombinant protein vitronectin, respectively [55,56]. However, homogeneous expansion of PSCs is still challenging to stabilize as 2D culture methods for propagation of PSCs are laborious, expensive, and require a high level of expertise. In general, 2D culture conditions favor the non-specific differentiation of PSCs.

The use of single-cell and multi-cell spheroids has proven to be an efficient system to optimize and overcome limitations associated with in vitro conventional systems. SCs are generally cultured in vitro under non-adherent conditions as spheres or adherent conditions in two-dimensional cultures or three-dimensional matrices. This method represents one of the simplest 3D culture techniques to achieve by forming multicellular aggregates, or spheroids, which allow 3D interactions with cells and the ECM in the absence of additional substrates [57]. A spheroid culture system provides a similar physicochemical structure that closely mimics the in vivo tissue counterparts by facilitating cell–cell and cell–matrix interaction, which play a significant role in various cellular mechanisms, subsequently maintaining the cellular properties [58]. These spheroids have been utilized with a wide range of adherent cell types, formed by spontaneous or forced aggregation techniques including hanging drop, rotating culture, or low-adhesion culture plates in suspension culture [59,60]. During spheroid formation, dispersed cells aggregate due to long-chain ECM fibers consisting of RGD (the tripeptide Arg-Gly-Asp) motifs that allow binding cell-surface integrin, and this leads to upregulated cadherin expression. Cadherin accumulates on the surface of the cell membrane, and the hemophilic cadherin–cadherin binding between neighboring cells allows for tightening connections between cells, and spheroids are formed [61,62]. Spheroids comprise highly proliferative, non-proliferative, and apoptotic cells with limited diffusion of oxygen and nutrients to the center of the spheroid, leading to an increasing hypoxic environment [63]. Due to their heterogeneous nature, spheroids have been more successfully employed to study complex 3D cell structures, cell differentiation, and cancer biology than homogenous cell proliferation [57]. However, long-term suspension culture of spheroids often results in aggregation of cells, leading to necrotic centers due to limited diffusion of nutrients and oxygen into and waste out of the aggregate. In contrast to ESCs, short-term spheroid culture has been employed to maintain and expand mesenchymal stem cells (MSCs) [57]. When subcultured back to 2D culture conditions, spheroid-grown MSCs displayed an undifferentiated morphology and enhanced differentiation potential via increased ECM deposition compared to adherent-grown MSCs [64]. MSCs cultured in 3D spheroids exhibited increased clonal growth and multipotency [65], altered miRNA expression, and increased acetylation in the promoter regions of pluripotency genes, OCT4, SOX2, and NANOG [66].

Organoids represent an essential bridge between traditional 2D cultures and in vivo mouse/human models. They are more physiologically relevant than monolayer culture models and are more amenable to manipulating niche components, signaling pathways, and genome editing than in vivo models. Organoids are classified into tissue and stem cell organoids, depending on the source from which they are formed. Stem cell organoids are generated from either ESCs, induced pluripotent stem cells (iPSCs), or primary stem cells such as neonatal tissue stem cells or tissue-resident adult stem cells. To date, several in vitro organoids have been established to resemble various tissues, including functional organoids for thyroid, pancreas, liver, stomach, intestine, vascularized cardiac patch, and cerebral [67].

A notable difference between organoids derived from primary tissue and ESCs/iPSCs is the presence of cell types other than the intended lineage in the latter. This is because the factors used for directed differentiation of ESCs/iPSCs are not entirely efficient in driving all the cells towards the lineage of choice; thus, many ectodermal and endodermal organoids, such as those of the intestine, stomach, and kidney, have limited presence of mesenchymal cell types [18,21,68]. In addition, the capability of self-renewing to grow in a near-physiological manner provides us with an excellent model system for a wide range of basic research and translational applications. A significant advantage of this system is the ability to greatly expand tissue-specific stem cells and their differentiated progeny from minimal amounts of starting material such as biopsies, facilitating detailed analyses of stem cell behavior, drug screening, disease modeling, and genetic screening. Indeed, intestinal organoids have already been used extensively to analyze stem cell behavior, identify niche components, model pathogen–epithelia interactions, gene editing, disease modeling, and orthotopic transplantation [3]. As organoids generated from ESCs, iPSCs, and fetal tissues faithfully retain the features of their original developmental stage, we can obtain detailed information of embryonic development in a dish as differentiation of the cells is systematically induced. It also delivers invaluable mechanistic insight into the development of stem cells and their niches while providing an opportunity to monitor their differentiation into mature functional lineages.

Another exciting application of the organoids is modeling host–microbe interactions. Some 3D tissue models have been applied to the study of microbial pathogenesis, such as hemolytic uremic syndrome caused by Shiga-toxin-producing *Escherichia coli* [69]. Studying bacterial and viral infection will allow a greater understanding of the pathogenic mechanisms and lead to better treatment strategies. Other studies have employed CRISPR/Cas9-mediated gene editing of healthy organoids to evaluate candidate gene function in tissue physiology and carcinogenesis directly. This advanced molecular technology system allowed the manipulation of specific genes enabling disease modeling and targeted gene therapy. Heart disease-associated human induced pluripotent stem cells (hiPSCs) have been derived from patients with cardiomyopathy, cardiomyopathy-associated Duchenne Muscular Dystrophy, familial long QT syndrome, prolonged QT interval, arrhythmia, hypertrophy, and myocardial infarction. In addition, through CRISPR/Cas9 technology, the possibility of creating hiPSCs with specific gene mutations or repairing known gene mutations to re-establish physiological cell function has become concrete. Moreover, this approach introduced serial mutations into healthy human colon organoids, converting them into cancer organoids capable of driving in vivo cancer formation following orthotopic or kidney capsule transplantation [69,70,71]. Patient-derived organoids also represent an essential resource for developing personalized treatment regimes. A wide variety of active drugs and small compounds can be screened for targeting candidate signaling pathways to design more effective drug regimens in conjunction with other relevant diagnostic and prognostic factors. Furthermore, in combination with 4D microscopy, organoids can be tracked over time to assess cancer stem cell behavior and viability in response to active drugs to predict patient outcomes.

## 4. Organoid Generation

Stem cells are primitive or “unspecialized” cells that have the potential to differentiate into many different and specialized cell types such as blood cells, muscle cells, bone cells, spleen cells, and other cells with specific functions [72,73]. Different methods exist for generating tissues from human pluripotent stem cells such as ESCs and iPSCs, and adult Stem Cells (aSCs) by mimicking the biochemical and physical cues of tissue development and homeostasis [74]. If provided with the proper microenvironment, ESCs and iPSCs can potentially differentiate into any tissue that arises from the three germ layers, but not the embryo, because they cannot give rise to the placenta and supporting tissues. Most tissues have multipotent stem cells, capable of producing a limited range of differentiated cell lineages appropriate to their location. This capacity to spontaneous differentiation into all three germ layers mimics embryonic development and promotes heterogeneous differentiation. In endoderm organoids generated from ESCs and iPSCs, TGF-β (transforming growth factor) signaling is stimulated to perform the definitive endoderm, differentiating into the corresponding embryonic viscera segment [75,76]. In organoids from the ectoderm, ESCs and iPSCs are induced to form “embryoid body”-like aggregates (PSC clusters) that are then driven to a neural or non-neural fate following ectodermal specification [20,77]. In organoids derived from the mesoderm, kidney organoids are generated by modulation of fibroblast growth factor (FGF) and GSK-3β signaling pathways in human iPSCs through a mesodermal intermediate stage [21].

aSCs are multipotent stem cells, able to differentiate in cell types closely related to the origin tissue. Creating conditions that mimic those of the natural stem niche can give rise to the tissue from which they were isolated. In addition, they play an important role in maintaining tissue homeostasis and tissue repair after injury. The first observation for tissue generation from aSCs was related to the growth of epidermal stem cells and large amounts of epithelium in vitro [78].

The advent of iPSC technology and the diversity of human aSC culture methods have made it possible, for the first time, to generate laboratory models specific to an individual [79]. Reprogramming other cells into iPSCs has become a routine laboratory procedure, but generating disease models from those cell lines remains challenging. Culture methods of iPSCs have been developed to mimic in vivo organ development in 3D, allowing more complex tissue structures and diverse cell types to be modeled simultaneously. In this methodology, human iPSCs are sequentially exposed to a course of differentiation cues to simulate the stages of a human developmental process. During this process, differentiated iPSCs aggregate to form first an organ bud and later organoids that truly mimic the mature organ structure, including multiple cell types and their interactions [2].

Human aSC-derived organoids have also emerged as an alternative organoid system that consists of a simpler structure. In contrast to the complicated process of iPSC reprogramming followed by differentiation to the required organ type, these organoids can be generated from biopsies isolated directly from the organ of interest or diseased patient tissue. However, the establishment of human aSC-derived organoids is limited by accessibility to the tissue and prior knowledge of the culture conditions for that tissue. At the same time, an iPSC line, once established from a patient, can be used to generate different tissue models without any time limit repeatedly (that is, beyond the patient’s lifespan) [80].

During organoid formation, many common factors are used to control spatially and temporally the self-renewal and differentiation of stem cells or assist self-organization. Growth factors or small molecules influence multiple signaling pathways critical in cell survival, proliferation, and self-renewal, often in a tissue-specific manner. The use of ESCs and iPSCs lines to generate organoids involves exposure to factors that promote germ-layer and tissue-specific patterning, incorporation into Matrigel medium to facilitate the development of 3D architecture and treatment with differentiation factors to produce desired organs. Matrigel, which is a heterogeneous and gelatinous protein mixture secreted by Engelbreth-Holm-Swarm (EHS) mouse sarcoma cells, provides a scaffold and additional supplementation of signaling cues via basement membrane ligands to support cell attachment and survival as well as organoid formation [81]. It comprises mainly adhesive proteins such as collagen, laminin, and heparin sulfate proteoglycans, which resemble the extracellular environment to provide structural support and ECM signals to the cells. Overall, matrigel provides a complex set of ECM signaling inputs and an appropriate mechanical context to organoids in vitro [82,83]. However, Matrigel has no defined composition and is animal-derived, representing a limitation to translation in clinical settings. As an alternative, essential signals from native ECMs can be incorporated into synthetic polymer matrices to produce designer ECMs like hydrogel generated from naturally occurring materials, such as fibrin [84], collagen [85], or hyaluronic acid [86] and synthetic hydrogels [87]. Many laboratories have used biochemically inert crosslinked hydrogels such as polyethylene glycol (PEG) or alginate to encapsulate cells in 3D [88]. These biomimetic scaffolds can be designed for specific organoid applications, and they can be constructed from either synthetic polymers (such as polyacrylamine and polyethylene glycol, PEG) or natural macromolecules (i.e., agarose or collagen) that can be used to make them permissive to biological processes. Indeed, engineering ECM with a material capable of supporting the spectrum of cell behaviors, critical for stem cells surviving, dividing, differentiating, and ultimately self-organizing into organoid-like structures, has involved considerable efforts, but further work is needed to identify materials with a full spectrum of chemical and physical properties necessary to support organoid growth and differentiation from stem cells.

Engineered matrices have been necessary for isolating the effects of mechanical cues on stem cell activity, independent of biochemical signals. Stiffness is a critical parameter influencing stem cell behavior and appears to be a determinant of the differentiation of MSCs towards different lineages [89]. Engineered matrices for organoid cultures might be designed with such physical considerations, including stiffness, matrix visco-elasticity, and degradability, which must be optimized for each specific organoid system. The efficiency of stem cell self-renewal and differentiation is correlated to ECM, and alterations in ECM composition are a hallmark of many diseases [90,91].

For scaffold-free techniques, cells are cultured in droplets of a defined culture medium hanging from a plate by gravity and surface tension [92]. Alternatively, the 3D structure of the organoids can also be established via “air–liquid interface”. In this case, cells are cultured on an MEF feeder layer or Matrigel initially submerged in a medium, which gradually evaporates and exposes the upper cell layers to the air to allow polarization and differentiation [93,94]. The initial culture conditions determine whether a cell (or cluster) will form an organoid by self-organization [95,96], and the size and number of EBs have also been shown to affect differentiation efficiency [97]. The size and shape of the initial cell aggregates are important starting conditions for organoids formation. Microwell structures or microfluidic devices usually achieve controlled cell aggregation; alternatively, the surface of cells, the cell membrane, can be modified to improve or initiate cell clustering [98]. There are different strategies for controlling spatial cell arrangement in vitro, from binding peptides proteins, nanoparticles, polymers, or bio-orthogonal chemical species [99,100], to cell surface binding of 3D DNA origami nanostructure that enables the programming of cell–cell adhesion [101]. In addition, it is possible to apply a magnetic field to cells magnetized by membrane-binding nanoparticles [102]. Beyond modifying cellular surfaces, a wealth of genetic engineering strategies is available to control the intrinsic properties of a cell [103,104].

Cells can be engineered by targeting components of pathways that control stem cell differentiation and key niche signals or by genome editing (i.e., CRISPR–Cas9 technology). The CRISPR system is much more flexible than existing techniques that use proteins such as transcription activator-like effectors (TALEs) and zinc-finger proteins. Although effective for targeting DNA in a sequence-specific manner, these systems utilize proteins that contain a DNA-binding domain (rather than nucleic acids) for their target specificity [105]. Genome editing could modulate the intrinsic response of cells in an organoid to external stimuli or induce specific differentiation drivers that could be employed for the terminal differentiation of cells. In 2013, it was shown that CRISPR could be applied in mouse and human intestinal organoids either to knock out a gene or to correct a disease-causing mutation cystic fibrosis transmembrane conductor receptor (CFTR) [106,107]. Furthermore, genetic bioengineering could be deployed to knock down relevant signaling pathways in specific cells to make them unresponsive to the corresponding stimulus, which could address the lack of spatial signaling control in organoids. Retinal organoids, derived from iPSCs from patients with retinitis pigmentosa, could also be gene-corrected to rescue defects in photoreceptor morphology and function [108]. In 2015, Freedman et al. showed that CRISPR could be used to model disease phenotypes in hPSC-derived organoids, creating an in vitro model for polycystic kidney disease [109]. Although the initial stages of differentiation appeared identical between PKD mutant and control organoids, PKD-mutant organoids behave differently when differentiated toward more mature kidney cells. Moreover, the tubular structures that develop in wild-type hPSC-derived organoids, the maturation of PKD organoids, results in large cystic structures, a condition present in PKD patients [109]. Therefore, gene editing in organoids can be utilized to elucidate signaling pathways responsible for disease development.

Stem cell behavior in vivo is highly regulated through the extrinsic biochemical and biophysical signals from specialized microenvironments. These microenvironments consist of a complex array of signaling mechanisms from niche support cells, the ECM, and mechanical forces, as well as systemic and physiochemical conditions such as oxygen and pH levels. The capacity to self-organize leading to reproducible and tightly regulated tissue architectures in vitro is a considerably variable process. Key chemical, physical, and spatial cues that guide the progress of self-organization in vivo may be lacking after cell reaggregation in vitro [110]. In these cases, the tools and techniques of engineering could facilitate the more robust formation and analysis of organoids. New engineering technologies such as microwell arrays, droplet-based microfluidics, 3D bioprinting with microscale or nanoscale topography, chemically programmed tissue assembly, and chemically defined ECMs mean that it is now feasible to engineer organoids in such a way as to precisely determine their initial size, composition, and spatial organization [87,88,111,112]. To provide shape-guided morphogenesis in vitro, technologies such as micro-manufacturing, 3D printing, and laser cutting could be used. For example, a poly-dimethylsiloxane (PDMS) stamp can be used to pattern a collagen scaffold, onto which organoid-derived cells can then be seeded, or laser-shaped matrices can be applied [113,114].

Another significant challenge in constructing an artificial scaffold in vitro is precisely replicating the presentation of signals to cells supplied in a precise spatial and temporal order. In traditional 3D cultures, cells are flooded with biochemical signals without any spatio-temporal control. This limitation can be addressed by 3D culture matrices that can release or present biomolecules under spatio-temporal control [115,116,117]. A possible approach to overcome this limitation is the delivery of morphogen gradients by microfluidic devices that can further induce controlled symmetry breaking. In 2016, Demers and colleagues developed a versatile microfluidic device capable of reproducing the spatial and temporal chemical in vivo environment during neural tube development [118]. Spatial control can be carried out by independent immobilization spatial-specific of different growth factors, promoting cell migration and differentiation within an agarose hydrogel by employing two-photon photochemistry [119]. Temporal control can be improved through photochemistry, which allows the design of matrices that locally release soluble chemical factors (or expose masked ones) in response to light stimulation [120,121].

## 5. From Cardiac Spheroids to Cardioids

The heart is a highly specialized organ that possesses a limited capacity for self-repair and regeneration after infarction or disease. Advances in the understanding of stem cell biology as well as advances in tissue engineering have provided an unlimited source of cells, particularly cardiomyocytes (CMs), for the development of functioning cardiac muscle capable of generating force and propagating electrical signals. Cardiac muscle is composed of several cell types. The main population is represented by CMs, but the endothelial cells and fibroblasts are the most abundant cell type when referring to numbers. Cardiac muscle tissue engineered for regeneration requires a significant resource of functional CMs that could be derived either from PSCs and/or cardiac stem cells (CSCs). Most of the current knowledge of cardiovascular biology and disease arises from the use of animal models, predominantly mice, rats, and pigs. Even though animal models have always represented a key informative tool to study heart disease, many aspects, such as the species difference in functional and biological properties, limit their translation to human cardiac disease and drug testing [122].

In the development of cardiac organoids, the current limitation on which several research groups are focusing their effort, consist in reproducing the vascular network of cardiac tissue. Current models do not yet achieve proper vascularization and efficient angiogenesis, conditions necessary for the delivery of nutrients and oxygen. The solution produced by techniques such as microfluidics, with which it is possible to control the size and shape of new cellular constructs, has allowed significant progress; however, several bottlenecks in reproducing all the necessary conditions of the heart structure in vitro are still unresolved. The known difficulty in deriving human CMs and their propagation in culture has hindered their use and instigated the use of alternative models. Human pluripotent stem cell-derived CMs represent a valuable resource for modeling the human heart, with significant advantages with respect to the study of toxic effects and efficacy of new drug therapies, as well as in regenerative therapy to treat cardiovascular diseases. One of the main limitations in the use of hPSC-derived CMs is their immature phenotype having contractile and other biological and physiological properties that differ from those of adult human CMs. CMs derived from hPSCs are typically generated through 3D aggregates (EBs) from 2D differentiation cultures using hanging drop methods, suspension cultures, or forced-aggregation method (Table 1). Usually, EBs are placed in matrix-covered plates for further differentiation, and the presence of contractile CMs will be assessed in the following days. Cardiac differentiation of EBs can be manipulated by the addition of cardiac growth factors, specific morphogenes, or by transgenic modifications to boost PSs cardiac commitment. CMs from cardiospheres revealed enhanced structural maturation with respect to CMs from 2D cultures [123]. Crucial factors that improve the efficiency of cardiac differentiation are the confluency in culture before differentiation, the size of the hPSC undifferentiated aggregates, and the size of EBs. Depending on confluency, the differentiation process is affected by the increase of cell-to-cell interactions and the associated paracrine factors. After 4–7 days of culture in suspension, EBs are plated onto gelatin- or Matrigel-coated dishes and cultured using a cardiomyogenic differentiation media. Despite that the beating foci of cardiomyocytes-derived from PSC occur within the adherent EBs layers, myogenic differentiation can also be observed in EBs cultivated the whole time in the form of floating EBs. Nevertheless, the growth conditions can considerably influence the microenvironment generated in culture and therefore also the differentiation efficiency. Specifically, myogenic differentiation of pluripotent cells can be enhanced by stage-specific application of key growth factors in defined media [124,125,126]. A variety of serum-free defined media for cardiac differentiation in EBs have been tested [126,127,128]. StemPro-34 is a serum-free media firstly used to culture hematopoietic progenitors. Nowadays, it has been also used for EB cardiac differentiation [124,127]. In the absence of serum to induce cardiogenesis, different growth factors implicated in normal cardiac development, including BMP4, activin-A, FGF2, Wnt agonists and antagonists, and vascular endothelial growth factor, were tested. Cardiomyogenesis has been shown to be sensitive to the glycogen synthase kinase-3 (GSK-3) inhibitor CHIR99021 concentration, while aggregate size did not prove to be a prevalent factor among culture platforms [126]. Successively, Yan and colleagues have demonstrated that cardiovascular spheroids derived from hPSCs and treated for 48 h with CHIR99021 express higher level of α-actinin and higher ratios of sarcomeric striations and Z-lines sarcomeres compared to 2D cultures [129]. 3D aggregates of hiPSC-derived cardiomyocytes (hiPSC-CMs) in comparison with 2D cultures have displayed down-regulation of the genes involved in glycolysis and lipid biosynthesis and increased expression of the genes involved in the mitochondrial oxidative phosphorylation system [130]. Light microscopy inspection of EB derived from hPSCs-derived CMs has revealed that the cells in the border of the EB are elongated with a rod-shaped morphology, with cross-striations of myofibrils in the cytoplasm. On the contrary, the CMs located at the center of EB have shown a round shape, no striations, but numerous lipid vacuoles and glycogen accumulation. Electron microscopy has shown that all EBs contain similar cells with ultrastructural features of CMs, surrounded by a monolayer of epithelial cells. Among the differentiated CMs at the center of EBs, it has been possible to visualize small cellular spaces containing collagen fibrils and cellular debris. The CMs nuclei located in the center of EBs showed a slightly irregular outline, while those located in the periphery were oval [131]. In addition, immunofluorescence staining has shown many different cardiac-specific markers, including troponin-T, α-actin, atrial myocyte-specific MLC-2a, ventricular myocyte-specific MLC-2v beating cells, and pacing cell-specific HCN4 [132]. CMs induction from PSCs has resulted in mixtures of ventricular-like, atrial-like, nodal-like cells, and pacemaker-like cells defined by intracellular electrophysiological measurements of action potentials [125]. Indeed, the phenotype of CMs-derived PSCs is characterized by a combination of sub-populations but also presenting a different degree of maturity. As reported in several studies, it is possible to guide differentiation towards atrial-like or to ventricular-like CMs by modulating the retinoic acid and Wnt signaling pathways [133,134,135].

In 2003, the identification and characterization of niches of endogenous CSCs in the adult mammalian heart had been associated with the expression of type III receptor tyrosine kinase c-kit (CD117 or SCFR-stem cell factor receptor) [136]. However, the adult heart contains a heterogeneous c-kit${}^{pos}$ cell population, most of which displays blood and endothelial lineage-commitment [137,138,139,140,141]. Recently, it was demonstrated that only a fraction of c-kit${}^{pos}$ cardiac cells, negative for CD45 and CD31, are enriched for multipotent CSCs [43,142,143]. These cells are distributed throughout the myocardium with the highest density in the atria and apex [139,144]. CSCs can be propagated over long-term culture and maintained in an undifferentiated, self-renewing, and stable state, without showing senescence or abnormal karyotype [145]. CSCs are clonogenic in vitro, and when grown in differentiation media for endothelial, smooth muscle, and cardyomyocyte lineages, acquire phenotypic characteristics of these different cell types.

When CSCs grow in suspension, they form spheres of hundreds of cells, similar to the pseudo-embryoid bodies created by the neural stem cells (neurospheres), which by analogy were named “cardiospheres” [136]. Cardiospheres are self-assembling, multicellular floating clusters that represent a distinctive feature of multipotent cells (Figure 2a,b). These cardiac-derived multicellular spheroids, are spontaneously formed by cardiac stem/progenitor cells and are cultured on a bacteriological dish or a poly(HEMA)-coated dish [43,146]. Immunostaining revealed that cardiospheres consist of c-kit${}^{pos}$ CSCs and supporting cells bound together by ECM proteins and connexins [136,147]. The architectural features of cardiospheres resemble those of in vivo stem cell niches, where stem cells are surrounded by supporting cells and linked by interactive ECM molecules [148]. Li et al. have shown that human cardiosphere-derived CSCs display the greatest myogenic differentiation potency in vitro compared with human BM-derived MSCs, adipose tissue-derived MSCs, and BM-derived mononuclear cells [148]. Marban and colleagues isolated a human cardiac progenitor cells (c-kit${}^{pos}$) population from endomyocardial biopsy composed by a different sub-population expressing endothelial (CD31 and CD34) and mesenchymal markers (CD90, CD105). Cardiospheres derived from this heterogeneous population expressed c-kit in the core and the mesenchymal marker CD105, CD31, CD133, and MDR-1 on their periphery. These cardiospheres also expressed Connexin43, NKX2.5, and desmin whereas in the periphery show cardiomyocyte-specific sarcomeric proteins (cTNI and αMHC). Therefore, such cardiospheres possess a core characterized by the expression of cardiac stem/progenitor markers while the cells located at the periphery seem to represent spontaneous differentiation of precursor cells into endothelial, mesenchymal, or cardiomyogenic lineages [149]. However, to boost CSC potential is indispensable to refocus the investigation on cloned Lin${}^{neg}$c-kit${}^{pos}$ CSC, since they represent the true pool of multipotent adult cardiac stem/progenitor cells. Indeed, we have demonstrated that the efficiency of cardiomyogenic induction using clonally derived CSCs is remarkably higher when compared to heterogenous freshly isolated CSC-enriched cardiac cells (Figure 2c) [43]. Cloned CSC showed robust self-renewing and serial sub-cloning capacity, genome stability, and multi-lineage cardiac cell differentiation potential [43]. Importantly, they generate cardiospheres at high frequency, which give rise to secondary and tertiary cardiospheres. When placed in laminin-coated plastic dishes with LIF-deprived basic differentiation medium for 14 days, the peripheral cells of the spheres expressed high levels of smooth muscle as well as endothelial and cardiac lineages. Nevertheless, the CM derived from cloned CSCs still reach a biochemical myogenic differentiation that is intermediate between fetal and neonatal cardiomyocytes [150]. These data have also been confirmed by RNASeq expression profile analysis. Comparison of transcriptome and miRNome between clonal CSCs, contracting cardiomyocytes obtained from CSC-derived cardiospheres (iCMs), and adult cardiomyocytes (aCMs) has shown that CSCs are cardiomyogenic since they activate the entire gene network and specific cardiomyo-miRs characteristic of the aCMs phenotype. Yet, in concordance with the immature phenotype of iCMs, the expression of these miRNAs is still significantly lower than in aCMs, falling between CSCs and aCMs [150].

Cardiosphere attachment on laminin or 0,1% gelatin plates alters cell shape and geometry, leading to cell flattening and changes in the cytoskeleton and nuclear shape. Cardiospheres show upregulation of many stem cell-relevant factors, such as c-kit, SOX2, Nanog, IGF-1, and Tert, which play an essential role in the growth and maintenance of the undifferentiated state [148]. Interestingly, the expression of HDAC2, an important histone deacetylase, is also upregulated in cardiospheres. The mechanism underlying this culture-acquired enrichment in stemness is unclear but may be related to the recapitulation of a stem cell niche microenvironment and HDAC2-mediated epigenetic modification. In addition, ECM and adhesion molecules, including laminin-β1, integrin-α2, and E-selectin, are also upregulated in cardiospheres [148]. As other spheres, also cardiospheres face a limited supply of oxygen and nutrients to the center of the spheroid.

Moreover, the work of Smith et al. has shown the feasibility of generating human and porcine cardiospheres and expanding stem cells from routine endomyocardial biopsy specimens. Cardiosphere-derived cells (CDCs) were isolated from percutaneous endomyocardial biopsies of the adult patients and examined in vitro for biophysical and cytochemical evidence of cardiogenic differentiation [151]. Human and porcine CDCs differentiated into electrically functional myocytes in vitro. Direct injection of human CDCs into the infarct border zone of SCID mice led to functional improvement and myocardial regeneration documented respectively by echocardiography and histology. CDCs transplantation induced both cardiomyogenesis and angiogenesis [151]. Other groups have investigated the ability of cells to secrete cytokines and growth factors (i.e., VEGF, HGF, and IGF) and the potential contribution of paracrine mechanisms to the beneficial and protective effects of cardiosphere-derived cells in vitro and in vivo [152]. Numerous studies have demonstrated that CDCs can engraft, differentiate, and improve cardiac function post-MI in mice [148,152], rats [153,154,155], and pigs [156,157]. While using the intracoronary route, CDC-treated pigs had similar values of EF to placebo-treated animals [156]; a slight improvement in the EF in post-MI pigs was obtained when CDCs were combined with βFGF [158].

## 6. Cardioids

The hiPSC-CMs are useful for traditional two-dimensional (2D) monoculture. Still, this model usually lacks a complete functional maturation [159], differentiation ability, and gene expression, which hampers their capacity to accurately predict human biology and pathophysiology in some instances [160]. Nevertheless, this gap could be overcome due to the possibility of generating hiPSC-CMs three-dimensional 3D structures [160,161,162,163]. Therefore, cardiac organoids, named as “Cardioids”, are generated from hiPSCs that self-organize into complex native-like organ [164] structures [165], representing faithfully many features of cardiac development stages (Figure 3). Cardiac organoids are usually prepared by inducing the aggregation of specific cells suspended in a medium or embedded in the 3D gel matrix. Recently, it was shown that hiPSC-derived EBs adherent to collagen-conjugated hydrogels are more successful in forming myocardium-like tissue [166]. Collagen-conjugated hydrogel has a stiffness similar to that of myocardial tissue, and probably for this, they can better stimulate cell proliferation and then differentiation into cardiomyocytes. However, there are some limitations related to matrix use. First, using a soft matrix cannot transmit mechanical signals that are insoluble during cardiomyocyte differentiation. Therefore, the difference in gel stiffness results in a difference in the degree of differentiation.

Artificially engineered heart tissues were successfully obtained through many bioengineering approaches using scaffolds, molds, cell hydrogel matrices, and 3D-printed biomaterials [160,167,168,169]. These approaches effectively measure contraction force, perform compound screens, and model structural muscle and arrhythmogenic disorders.

Nevertheless, many previous methods [165,170,171,172,173] do not recapitulate cardiac-specific self-organization to acquire an in vivo-like structure [28]. To this aim, using a high-throughput approach, it was demonstrated that WNT and BMP, the central regulators of the cardiac specification [174], drive chamber-like self-organization [28]. Moreover, the inhibition of WNT signaling at the cardiac mesoderm stage seems crucial for CM specification but is not necessary for cardioid self-organization. Recent work uses a protocol based on three sequential modulation steps of the WNT pathway (activation/inhibition/activation) at specific time points on suspension EBs to drive the production of significant heart-like structures in terms of organization, functionality, cardiac cell type complexity, ECM composition, and vascularization [175]. The gene expression profile of cardioids showed upregulation of cardiac-specific genes and downregulation of WNT signaling at day 20 differentiation and significantly higher gene expression of transforming growth factor β (TGF-β) signaling (such as TGFβ1, TGFβ2, TGFβ3, TGFβR1, and TGFβR2) and cardiac-specific genes (MYL4, MYH7, and NKX2.5 [176]). Comparison of transcriptomes of 2D iPSC-CMs, 3D iPSC-CMs, human cardioids (hCOs) RNA-seq datasets of healthy human myocardium derived from fetal atria, fetal ventricles, and adult ventricles showed that hCOs best summarize cellular diversity and share the highest transcriptomic similarity to fetal myocardium, presenting fibroblast specific ECM organization, endothelial cell vascularization, and early immune cell regulation [28].

In cardioids, the patterning and morphogenesis of CM and endothelial cell lineages are controlled by the combination of specific cardiac and endothelial growth factors. Specifically, at the early time point of mesodermal differentiation, WNT and ACTIVIN were added to differentiation media, while the next step requires adding VEGF to move on both specification and patterning of the EC layer in cardiac mesoderm [28]. Therefore, cardioids are a promising system to study the underlying mechanisms of CM and EC patterning and crosstalk in the context of a beating chamber. Moreover, cardiac organoids represent a booster to investigate mechanisms of human cardiogenesis. However, the self-assembly process used in most organoid procedures is still a limiting factor for a consistent generation of cardiovascular tissues. Mostly, this process is a random method resulting in heterogeneous organoids regarding cell composition, size, and shape [95]. Nonetheless, the application of hiPSC-derived cardiac organoids to disease modeling shows countless advantages in clinical medicine, contributing to the investigation of a large variety of phenotypes and a robust technology applicable in drug development and screening [160,177,178,179,180,181,182].

Recently, cardiac organoids that incorporate an oxygen-diffusion gradient and that are stimulated by the neurotransmitter norepinephrine resemble the structure of the human heart after myocardial infarction (by mimicking the infarct, border, and remote zones) recapitulating transcriptomic, structural, and functional hallmarks of myocardial infarction [165]. Furthemore, these organoids can model hypoxia-enhanced doxorubicin cardiotoxicity [165]. The authors utilized non-adhesive agarose hydrogel molds to make hiPSC-CMs and non-myocyte mixtures used to form human cardioids. The human cardioids were placed into a hypoxic chamber and were treated with 1 μM noradrenaline to create an apoptotic gradient, which simulates the environment of myocardial infarction in vitro. The transcriptomic data and functional analyses showed that infarcted organoids recapitulate key aspects of metabolism in the human infarcted myocardium. However, the absence of inflammatory cells or the level of maturation of hiPSC-CMs obtained in vitro implies the lack of perfect simulation of heart failure in the infarct organoids [183]. Changes in transcriptome level explain the extent to which infarcted organoids could model responses of human cardiac tissue after infarction. From this perspective, iPSC-based systems are likely to be very helpful to model diseases across many adult tissues, including the heart [74,184].

Another approach has been the generation of 3D cardioids from mouse embryonic stem cell-derived EBs using the substrate laminin-entactin (LN/ET), which includes components of the ECM in connective tissue, and exogenous FGF4 for induction of CM proliferation and cardiac chamber formation. Interestingly, when EBs were generated without the LN/ET complex, they failed to undergo morphological changes suggesting that the LN/ET complex promotes an extracellular environment for heart development. They also investigated the presence of cardiac muscle-specific structures, finding the appearance of intercalated discs, sarcomere structures with Z-bands, mitochondria, and desmosomes, Purkinje cells, and T-tubule [185]. Among all, the intercalated disc is involved in the coordination of muscle contraction. Therefore, these cardioids might possess contractile cardiac muscle cell properties. Thus, this heart organoid culture system provides a valid method to significantly improve regenerative medicine, study congenital heart diseases, and screen potentially dangerous drugs that cause cardiac defects.

A human in vitro model of acute cryoinjury has been also developed, which seems to be physiologically representative of the native immature human heart and exhibits an endogenous regenerative response [178]. A different method includes generating a 3D microtissue system composed of the primary cardiac cell line, CMs, cardiac endothelial cells, and cardiac fibroblasts derived entirely from hiPSCs [171]. This system seems to promote the crosstalk between different cell lines improving cardiomyocytes maturation. Specifically, they identified some key mechanisms in the tri-cellular interactions that enhance CM maturation. One mechanism observed in tri-cellular interactions that enhance CM maturation includes the increased cAMP levels in CMs positively affecting the assembly of CX43 gap junctions [171]. The silencing of CX43 significantly reduced the structural organization of sarcomeres and the contraction duration [171]. Other factors such as paracrine effects or cell-extracellular matrix interactions may also contribute to CM specification and maturation.

Taken together, the main purpose of all the experimental procedures is to obtain fully mature hiPSC-CMs to mimic faithfully in vivo heart structures. Combining genomic editing technology and hCOs, it is possible to precisely modify and correct each mutation and realize innovative and personalized therapeutic platforms for disease modeling. Long and colleagues have used hCOs to show that dystrophin mutations impaired cardiac contractility and sensitivity to calcium concentration and that correcting mutations in the X-linked dystrophin gene by myoediting contractile dysfunction was partially-to-fully restored [186]. hCOs can be used to study more electrophysiological phenomena, such as conduction and reentry, in arrhythmogenic syndromes, such as short QT syndrome, exploiting their ability to produce both spontaneous and induced action potentials with a higher conduction velocity [187].

The development of 3D stamping and bioprinting techniques also provides scaffolds to generate heart-on-a-chip. Engineered heart tissues and heart-on-a-chip methods allowed modeling of specific cardiac diseases, including Barth syndrome-associated cardiomyopathy [188], Duchenne muscular dystrophy [186], and primary hypertension-induced left ventricular hypertrophy [169]. Compared to standard 2D culture formats, engineered 3D heart tissues improve CM maturity and exhibit a more physiological 3D muscle environment [189], representing the new frontiers to study human cardiac regeneration in vitro.

Based on the consideration that the adult heart contains a niche of endogenous stem cells [190,191,192], these could be considered a potential source for organoid generation. Cardiac organoids can be generated by isolating adult stem cells from small biopsy specimens. CSCs can release their own cardiac morphogens promoting the formation of structures of greatly increased complexity and finely organized architecture. CSCs have the potential to differentiate into many or all the cell types present in the heart and acquire anatomic morphology such as chamber organization and atrioventricular specification. Therefore, cardioids generated from CSCs may provide a basis for the study of congenital and age-related cardiovascular disease [142,193,194,195,196,197,198].

## 7. Future Directions

The findings discussed here illustrate that organoid technology is one of the most promising three-dimensional (3D) culture innovations in biological and medical research. Organoids display physiological functions specific to that organ and present cell organization similar to that of the organ itself [199]. However, organoids recapitulate some but not all aspects of the tissue of origin. Only a fraction of the in vivo cellular and physical environment is recapitulated. Mimicking the spatial and functional complexity observed in vivo is indeed a defined, yet not readily achievable, target to reach. Different approaches are being followed, from co-culture systems that combine different cell types using engineering approaches to mixing different types of already pre-patterned and differentiated structures to generate multiplex organ tissues. Organoid culture often requires Matrigel or another animal-based matrix extract to support cells to aggregate into 3D structures. The composition of these extracts can change between batches which may affect the reproducibility of experiments. In addition, they may transfer pathogens and are potentially immunogenic when transplanted to humans, limiting the application of organoids in a clinical transplantation protocol. This may be resolved by using clinical-grade collagen or a synthetic polyethylene glycol-based gel [87,200].

Organoids potentially provide alternative cellular sources for cell therapy transplantation, revolutionizing the future treatment of several chronic diseases or extending the time required for an organ transplant. They contribute to reducing the use of animal models and costs in the pharmaceutical industry. Despite that, these approaches tend to be expensive, work-intensive, and not readily scalable. A cardioids model, as a supplement to a preclinical model, can be used to simulate pathological processes and heart development effectively and to detect toxicity and side effects of drugs. Still, it has not yet wholly replaced animal models. Significantly, an important limitation of the hCOs is their inability to include all the cells found in the in vivo heart. hCOs, due to the lack of inflammatory cells and the use of immature hiPSC-CMs, cannot highlight the immune system’s role in myocardial infarction, heart failure, or other cardiac diseases. Their ability to recapitulate heart development is still limited compared to other models, such as mice, even though they have the significant advantage of being human in origin rather than a surrogate animal model [201]. There is large room for improvement in the technology, particularly in better recapitulating morphological and anatomical features and inducing the formation of effective vascular networks that can provide nutrients [202]. A fundamental limitation of many organoid systems is a lack of a functional vascular network to facilitate the exchange of nutrients and waste material removal, as they rely solely on diffusion [203]. Innovative co-culture methods can better simulate the interaction between multiple cell types, which will be helpful to constructing specific-chambered hCOs containing ventricle-like and atrial-like structures and functions in the future.

The real possibility to manipulate the expression of a specific target gene through the genome editing strategy makes organoids a suitable preclinical tool to approximate the therapeutic efficiency of gene editing.

Moreover, the organoid-on-a-chip platform is an innovative technology to fabricate 3D organ models, which may bridge the gap between monolayer cell cultures and animal models. The artificial assembly of multiple organoids or the combination of organoids with cells from different tissue lineages becomes essential for analyzing other parameters that may change under specific conditions [164]. This approach could replicate the definite functions at the multiorgan level and the complex processes of drug metabolism and reaction [204,205].

High-fidelity hCOs could be valuable in understanding human physiology, cardiac development, specific drug testing, disease modeling, and drug discovery. Therefore, in the last few years, 3D models have become increasingly more sensitive and are used to study many diseases such as neurogenerative diseases, cancer, and cardiomyopathy. Furthermore, 3D models in regenerative medicine are promptly developing and offer an unprecedented approach to personalized medicine. Both spheroids and organoids models are not only closer representations of in vivo tissues than 2D cell cultures but can also efficiently recapitulate human-specific biology in a dish and shed light on biological mechanisms, pathogenesis, and disease treatment. However, the current technologies for spheroid and organoid generation are limited by the inability to replicate the complex cell–cell interactions, cellular diversity, and microenvironment cues of tissues in vivo and lack of reproducibility. Further studies are required to refine the technology to go beyond and definitively abandon animal models to investigate critical cellular events in biology. Bioengineering strategies may provide new directions to overcome this issue.

## Figures and Tables

**Figure 1 ijms-22-13180-f001:**
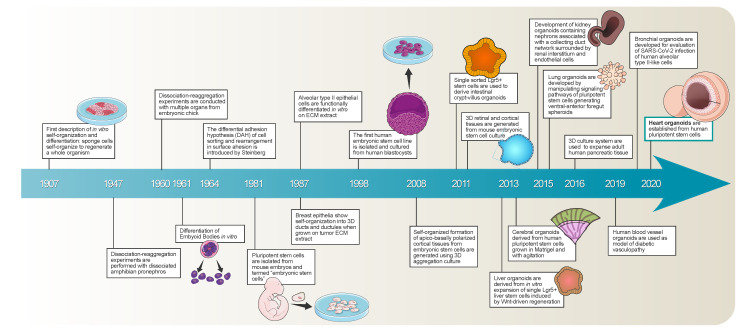
Milestones timeline of the methodologies for 3D cell cultures and organoids’ generation. Since the beginning of the 20th century, several different methodological breakthroughs have been established in biology and clinical translation leading to the current achievements, challenges, and potential applications of organoids.

**Figure 2 ijms-22-13180-f002:**
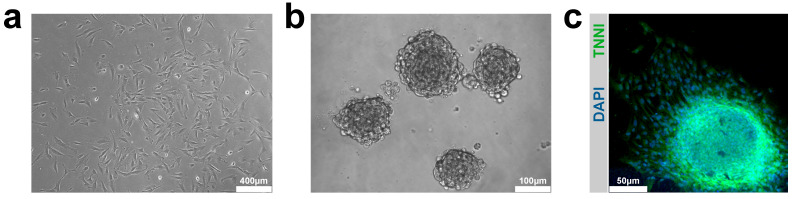
(**a**,**b**) Representative light microscopy image of monolayer culture and cardiospheres derived from c-kit${}^{pos}$/CD45${}^{neg}$/CD31${}^{neg}$ hCSCs. Scale bar = 400 μm and 100 μm. (**c**) Representative confocal microscopy images of cardiospheres derived-c-kit${}^{pos}$/CD45${}^{neg}$/CD31${}^{neg}$ hCSCs plated in cardiac differentiation media efficiently commit to cardiomyogenic cell lineages (TNNI, green). Nuclei are stained in blue (DAPI). Scale bar = 50 μm. Adapted from Scalise M. et al. [139].

**Figure 3 ijms-22-13180-f003:**
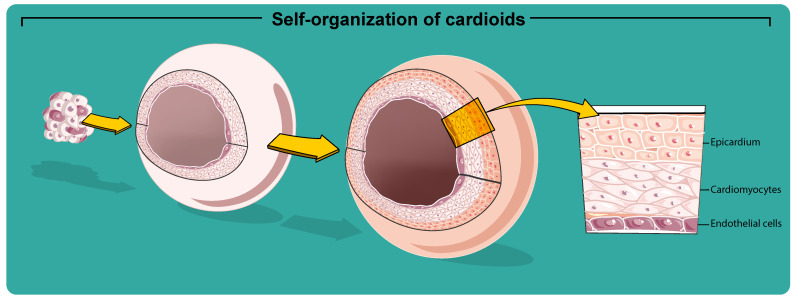
Cardioids recapitulate intrinsic self-organization and arrangement to form cardiac architecture and chamber-like structures growing from pluripotent and cardiac stem cells.

**Table 1 ijms-22-13180-t001:** Main methodologies of cardiosphere derivation with their characteristics.

Methodology	Growth Characteristics	Advantages	Disvantages
Suspension cultures	Uncontrolled spheroid size, non-uniform spheroid shape, variations in cell number	Simple, cheap, quick to scale up	Low efficiency
Hanging drops	Fast spheroid formation, well-controlled and uniform spheroid size	Simple, no special equipment is required	Laborious task, time-consuming, instable, difficult long-term cultures, low throughput, low efficiency
Forced aggregation	Rapid cell aggregation, well-controlled and uniform spheroid size	Increased efficiency, high throughput	More complex, undefined potential effects to cells, special equipment and culture conditions are required

## Data Availability

No new data were created or analyzed in this study, so sharing is not applicable to this article.

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
