# Peer review of "From Spheroids to Organoids: The Next Generation of Model Systems of Human Cardiac Regeneration in a Dish"

_ijms, 2021, doi:10.3390/ijms222413180_

Round 1

Reviewer 1 Report

The manuscript is a comprehensive detailed description of Organoid technology, its evolution over the years, and its utilization towards human cardiac organoid modeling. The review does a great job discussing the principles behind the development of organoid technology and the limitations of its utilization. It will be a valuable review for anyone dabbling in the field of cardiac organoids and organoid cultures. 

Author Response

Response to Reviewer 1 Comments

The manuscript is a comprehensive detailed description of Organoid technology, its evolution over the years, and its utilization towards human cardiac organoid modeling. The review does a great job discussing the principles behind the development of organoid technology and the limitations of its utilization. It will be a valuable review for anyone dabbling in the field of cardiac organoids and organoid cultures.

Response: We sincerely thank the reviewer for her/his complimentary comments and for the overall positive evaluation of our article.

Reviewer 2 Report

In the current literature review article authors focused mainly the organoids developed from the cardiac tissues and its use in understanding the complications associated with heart diseases. They discussed the evolving history of organoid technology and compared it with 2 dimensional (2D)  culture system and highlighted the benefits of use of organoids. Authors thoroughly described the alternate ways to generate cardiac spheroids and then in to cardioids and finally the future utility of this technology in the translational as well as basic research.

Comment

  1. Line #50: In vivo or ex vivo?
  2. Line #66 Matrigel: It is not an extracellular matrix (ECM) component but commercially developed product of mixture of various ECM components
  3. It could be interesting to present in tabular or graphical form of research involved in discovering organoids chronologically under section 2
  4. Organoids generated from various tissues from humans and methods used to develop organoids could be depicted in the graphical format
  5. Line #426 correct the sentence
  6. In section 5, list the methods of cardiospheres derivation methodology with their media composition, growth characteristics, advantages and disadvantages
  7. Line #673 correct the sentence
  8. It could be interesting to add sub-sections under section 6 and elaborate the methodology/approaches of developing cardioids and its use in various applications

Author Response

Response to Reviewer 2 Comments

In the current literature review article authors focused mainly the organoids developed from the cardiac tissues and its use in understanding the complications associated with heart diseases. They discussed the evolving history of organoid technology and compared it with 2 dimensional (2D)  culture system and highlighted the benefits of use of organoids. Authors thoroughly described the alternate ways to generate cardiac spheroids and then in to cardioids and finally the future utility of this technology in the translational as well as basic research.

Response: We sincerely thank the reviewer for her/his complimentary comments and for the overall positive evaluation of our article. We believe to have satisfactorily addressed all her/his comments as detailed below.

Point 1: Line #50: In vivo or ex vivo? 

Response 1: In vivo

Point 2: Line #66 Matrigel: It is not an extracellular matrix (ECM) component but commercially developed product of mixture of various ECM components 

Response 2: Thanks. Correction made.

Point 3: It could be interesting to present in tabular or graphical form of research involved in discovering organoids chronologically under section 2 

Response 3: Thanks. A new figure (now Figure 1) has been added to the manuscript.

Point 4: Organoids generated from various tissues from humans and methods used to develop organoids could be depicted in the graphical format – Done

Response 4: Thanks. A new figure (now Figure 1) has been added to the manuscript.

Point 5: Line #426 correct the sentence – Done

Response 5: Thanks. Correction made.

Point 6: In section 5, list the methods of cardiospheres derivation methodology with their media composition, growth characteristics, advantages and disadvantages 

Response 6: Thanks. A new table (now Table 1) has been added to the manuscript.

Point 7: Line #673 correct the sentence

Response 7: Thanks. Correction made.

Point 8: It could be interesting to add sub-sections under section 6 and elaborate the methodology/approaches of developing cardioids and its use in various applications

Response 8: We thank the reviewer for her/his comments but we hope that s/he understands our belief that despite adding subsections could be a valuable formal editing, the section as a whole reads fine as it is listing all the topics that the reviewer refers to.

Reviewer 3 Report

A well-written article with details. Please make sure all the short forms are explained when mentioned for the first time in the article (as for example  SARS-CoV-2). 

In the introduction, the authors should mention the highlight or the rationale of the review. They already mentioned it but having a brief overview in the introduction helps the readers to find out what they expect from reading the whole article. 

Author Response

Response to Reviewer 3 Comments

We sincerely thank the reviewer for her/his complimentary comments and for the overall positive evaluation of our article. We believe to have satisfactorily addressed all her/his comments as detailed below.

Point 1: Please make sure all the short forms are explained when mentioned for the first time in the article (as for example SARS-CoV-2).

Response 1: Thanks. Corrections made.

Point 2: In the introduction, the authors should mention the highlight or the rationale of the review. They already mentioned it but having a brief overview in the introduction helps the readers to find out what they expect from reading the whole article.

Response 2: Thanks. We have added the rationale of the review at the bottom of the first paragraph.
